# Simultaneous Analysis of Hydroquinone, Arbutin, and Ascorbyl Glucoside Using a Nanocomposite of Ag@AgCl Nanoparticles, Ag_2_S Nanoparticles, Multiwall Carbon Nanotubes, and Chitosan

**DOI:** 10.3390/nano10081583

**Published:** 2020-08-12

**Authors:** Nutthaya Butwong, Thidarat Kunawong, John H. T. Luong

**Affiliations:** 1Applied Chemistry Department, Faculty of Sciences and Liberal Arts, Rajamangala University of Technology Isan, 744, Suranarai Rd., Nakhon Ratchasima 30000, Thailand; thidarat.ku@gmail.com; 2School of Chemistry, University College Cork, T12 YN60 Cork, Ireland; luongprof@gmail.com

**Keywords:** ascorbyl glucoside, electrochemical sensor, hydroquinone derivative, silver-silver chloride nanoparticles, silver sulfide nanoparticles

## Abstract

A nanocomposite comprising Ag nanoparticles on AgCl/Ag_2_S nanoparticles was decorated on multi-walled carbon nanotubes and used to modify a glassy carbon electrode. Chitosan was also formulated in the nanocomposite to stabilize Ag_2_S nanoparticles and interact strongly with the glucose moiety of arbutin (AR) and ascorbyl glucoside (AA2G), two important ingredients in whitening lotion products. The modified electrode was characterized by Fourier transform infrared spectroscopy (FTIR), transmission electron microscopy (TEM), and scanning electron microscopy (SEM) and cyclic voltammetry and used for the simultaneous analysis of hydroquinone (HQ), AR, and AA2G. The electrode showed excellent electrocatalysis towards the analytes by shifting the anodic peak potential to a negative direction with ≈5-fold higher current. The sensor displayed a linearity of 0.91–27.2 μM for HQ, 0.73–14.7 μM for AR, and 1.18–11.8 μM for AA2G, without cross-interference. A detection limit was 0.4 μM for HQ, 0.1 μM for AR, and 0.25 μM for AA2G. The sensor was applied to determine HQ, AR, and AA2G spiked in the whitening lotion sample with excellent recovery. The measured concentration of each analyte was comparable to that of the high performance liquid chromatographic (HPLC) method.

## 1. Introduction

Melanin is synthesized from tyrosine by tyrosinase to form two eumelanins (brown to black) or one pheomelanin (yellow to red) in the absence or presence of cysteine, respectively. Thus, commercial skin lightening products are formulated with a mixture of tyrosinase inhibitors including ascorbyl glucoside (AA2G) and hydroquinone-*O*-β-D-glucopyranoside (β-arbutin, AR). Popular AR is widely used in the beauty industry as depigmenting cosmetics and anti-sunburn of human skin. The cosmetic effect of AR, a glycosidic form of hydroquinone, is weaker than hydroquinone but exhibits lower cytotoxicity, nephrotoxicity, and genotoxicity. AR undergoes hydrolysis to release hydroquinone (HQ) when it is subject to high temperature, ultraviolet (UV) radiation, dilute acid, or human skin bacteria [1]. Both kojic acid and alpha hydroxy acid are also allowed in cosmetics in Europe, but hydroquinone and tretinoin are prohibited. Thus, monitoring cosmetic ingredients with hydroquinone and ascorbic derivatives is essential to ensure product quality and consumer safety [2].

High-performance liquid chromatography (HPLC) separates and analyzes multi-components of lightening products [3]. In contrast, a standalone electrochemical sensor is often used to detect a single or two target analytes with different electroactivities such as AR and vitamin C [4]. Sensitive detection of some analytes is also not achieved due to their low electroactivity, e.g., glucose, AA2G, or AR unless the electrode surface is modified with a recognition molecule [5], nanoparticles [4], carbon nanotubes, etc. Differential pulse voltammetry (DPV) serves as a sensitive method for separating and detecting dopamine -uric acid [6] and or a mixture of HQ, catechol, and resorcinol [7].

Electrodes can be modified with metal nanoparticles to accelerate electron captor and electron transfer to impart detection sensitivity [8]. Ag_2_S/Ag nanoparticles are synthesized by the chemical deposition from an aqueous solution of Ag^+^, S^2−^, and reducing agents [8]. Heterostructures of AgNPs and Ag_2_S nanofibers or an Ag film with Ag_2_S nanoparticles are considered as novel sensing materials for biosensors and optoelectronic devices [9]. Albeit Ag/AgCl [10] or Ag/Ag_2_S [8] nanoparticles are widely used as a photocatalyst, the combined Ag_2_S/AgCl NPs and AgNPs have not been investigated. Moreover, both AgCl [11] and Ag_2_S nanoparticles [12] were found to catalyze the reduction of Ag^+^ ions to Ag^0^ under UV/visible light, which can provide the stability of AgNPs on their particles. Multiwall carbon nanotubes (CNTs) decorated Ag_2_S, AgCl, and AgNPs result in an enormous sensing surface area. Chitosan can be deposited on different substrates [13] and plays an important dual role as it interacts strongly with the glucose moiety of AR or AA2G [14] and stabilizes Ag_2_S NPs [15], two prerequisites for sensitive detection of these two target analytes by DPV.

This work unravels the electrochemical properties of a glassy carbon electrode modified by a composite of Ag@AgCl, Ag_2_S, CNTs, and chitosan for the simultaneous analysis of HQ, AR, and AA2G. To our knowledge, there has been no attempt for the simultaneous analysis of HQ, AR, and AA2G by electrochemical sensing. The modified electrode with a high active surface area is characterized and evaluated for its analytical performance of HQ, AR, and AA2G, three popular ingredients in a commercial whitening lotion product. The synergistic catalytic effect of Ag@AgCl, Ag_2_S, chitosan, and CNTs can be attributed to a negative shift of the oxidation potential of the analytes.

## 2. Materials and Methods

### 2.1. Reagents and Materials

Silver nitrate is obtained from Riedel-de Haën (Seelze, Germany) while multiwalled carbon nanotubes (CNTs, 20–40 nm diameter) were purchased from TCI Chemicals (Tokyo, Japan). Sodium sulfide is a product of BDH and the remaining chemicals were purchased from Sigma-Aldrich (St. Louis, MO, USA). All the chemicals were of analytical grade and used as received. Aqueous solutions were prepared by deionized water (18.2 MΩ⋅cm). Different pH-phosphate buffers (0.1 M) were prepared from 0.1 M NaH_2_PO_4_ and 0.1 M Na_2_HPO_4_ with the pH adjusted by 1 M H_3_PO_4_ or 1 M NaOH. HPLC-grade methanol was obtained from Thermo Fisher Scientific (Bangkok, Thailand).

### 2.2. Synthesis of Chitosan Stabilized Ag_2_S NPs

The colloidal Ag_2_S stabilized in chitosan was synthesized using a similar procedure used in previous work with some modification [15]. The chitosan suspension was prepared by solubilizing chitosan (1.0 g) in acetic acid (50 mL, 1.0 wt%) solution. AgNO_3_ (5 mL, 10 mM) was then added immediately into the suspension under constant stirring for 30 min. Then, 5.00 mL of 8 mg⋅mL^−1^ freshly prepared Na_2_S·xH_2_O was quickly added to the resulting Ag-chitosan suspension with stirring for 90 min. The resulting Ag_2_S NPs were kept at 4 °C.

### 2.3. Synthesis of the Ag@AgCl/Ag_2_S/CNTs on the Glassy Carbon Electrode

A glassy carbon electrode (GCE) was polished with alumina powder (0.5 µm and 0.03 µm), washed with DI water, ethanol, and DI water again and dried at room temperature. CNTs (100 mg of) were then dispersed in a mixture of concentrated H_2_SO_4_ and HNO_3_ with a volume ratio of 3:1 (50 mL) and sonicated for 6 h. After extensive washing, the neutral filtrate was dried at 100 °C under vacuum. The resulting carboxyl CNTs (1 mg) were dispersed into 1.00 mL DMF in an ultrasonic bath for 30 min to form a homogeneous suspension. Then, 3.0 μL of the 0.1% (*w/v*) CNTs in DMF was dropped onto the GCE surface and left to dry for 3 h under ambient conditions. For the electrodeposition of Ag_2_S-chitosan nanoparticles on CNTs, the CNTs/GCE was immersed into the Ag_2_S colloidal solution with the applied potential from −1.0 V to 0.1 V at a scan rate of 20 mV⋅s^−1^ for 15 cycles. Subsequently, AgNPs/AgClNPs (Ag@AgCl) were deposited on the Ag_2_S/CNTs/GCE by applying a potential range of −0.8 to 1.35 V (Ag/AgCl) at a scan rate of 20 mV⋅s^−1^ on the Ag_2_S/CNTs/GCE for 15 cycles in 5.0 mL of a mixture of 1% (*w/v*) ascorbic acid and 0.1 M KCl containing 0.5 mM AgNO_3_. The resulting electrodes were thoroughly washed with DI water and dried at ambient temperature, designated as the Ag@AgCl/Ag_2_S/CNTs/GCE.

### 2.4. Characterization

Electrochemical measurements were performed using an Autolab PGSTAT204 potentiostat (Metrohm Autolab B.V., Utrecht, Netherlands) operated with the NOVA 2.1 software. A three-electrode electrochemical cell consisted of a modified or bare GC as a working electrode, an Ag/AgCl (vs. Ag/AgCl, 3 M KCl) as the reference, and a platinum wire as the auxiliary electrode. The impedance measurements were conducted in 0.1 M KCl containing a redox couple (5.0 mM of K_3_[Fe(CN)_6_]/K_4_[Fe(CN)_6_]) with the frequency from 1.0 × 10^−3^ to 1.0 × 10^5^ Hz and signal amplitude of 5.0 mV. All pH values were measured with a digital pH meter (model 1230, Orion, Thermo Fisher Scientific, Waltham, MA, USA). Zetasizer Nano ZS (Malvern, MA, USA) was used for the characterization of Ag_2_S-chitosan. The UV-2450 UV-vis spectrophotometer (Shimadzu, Kyoto, Japan) and the PerkinElmer Spectrum 100 FT-IR spectrometer (Shelton, CT, USA) were used for study optical and functional groups of nanoparticles, respectively. The modified electrode was studied by the ZEISS GeminiSEM–field emission scanning electron microscope (SEM) equipped with an Oxford Instruments X-act EDX system (Oberkochen, Germany) and the transmission electron microscope (TEM) FEI Tecnai G^2^ 2 S-TWIN (Hillsboro, OR, USA). For SEM, TEM, and FTIR characterization, the modified electrode was sonicated in 1.0 mL of ethanol to get the nanomaterials on GCE to characterize by SEM, TEM, and FTIR.

### 2.5. Sample Preparation

A commercial whitening lotion (Nakhon Ratchasima, Thailand) sample was used as an example to demonstrate the sensor’s applicability. The whitening lotion sample (1.0 mL) was added to 100 mL of 100 mM of PBS, pH 7, and sonicated for 30 min. The suspension (10 μL) was diluted with PBS to 5.00 mL and used for the detection by DPV. For the HPLC method, 1.0 mL of sample was diluted to 25 mL of 50 mM of PBS, pH 2.5, and sonicated for 30 min. The suspension was filtered by a nylon membrane with a pore size of 0.45 μm. Quantitative analysis was performed by the standard addition method to eliminate the sample matrix effects. The analyte standard solution was spiked into the whitening lotion sample to evaluate the sensor performances including sample recovery, detection precision, and accuracy.

### 2.6. Electrochemical Measurements

The Ag@AgCl/Ag_2_S/CNTs/GCE was connected to the respective terminals of the Autolab electrochemical analyzer for measuring the voltammetric response. In all experiments, 5.0 mL of the supporting electrolyte solution was placed in the electrochemical cell. Cyclic voltammetric (CV) experiments were cycled from −0.8 V to +1.0 V with a scan rate of 100 mV⋅s^−1^. DPV responses were recorded between −0.1 and +1.0 V for different concentrations of HQ, AR, and AA2G standard solutions. The oxidation peak of HQ at +0.05 V, AR at +0.5 V, and AA2G at +0.7 V obtained in DPV were used for quantification. All measurements were carried out at room temperature (25 ± 2 °C). The analyte quantitation was obtained by measuring the anodic peak current and the standard addition method was used to evaluate the analyte contents in real samples. The CV experiments were performed in the range between −0.5 to 1.0 V at different scan rates (10 to 210 mV s^−1^) in 0.1 M phosphate buffer, pH 7. DPV was performed at 10 mV⋅s^−1^; pulse amplitude 50 mV; pulse width 0.05 s; sample width 0.0167 s; pulse period 0.2 s; quiet time 2 s.

### 2.7. High-Performance Liquid Chromatography Measurements

The lotion sample was analyzed using a Shimadzu Prominence LC-20A series HPLC system (Shimadzu Co., Kyoto, Japan), comprising an LC-20AT pump, CTO-20A column oven, SIL-20AC autosampler, and SPD-M20A PDA detector. The acquired chromatographic data were converted and processed by LC solution software (Version 1.24, SP1; Shimadzu, Kyoto, Japan). The column used for the separation of the analytes was an XBridge C18 3.5 um, 4.6 × 100 mm HPLC column (Waters Corp., Milford, MA, USA), maintained at 40 °C. The mobile phase consisted of the MeOH: 50 mM phosphate buffer (pH 2.5) at a ratio of 4:96 with a flow rate of 0.5 mLmin^−1^. All analyte absorbances were detected at 280 nm with an injection volume of 20 μL.

## 3. Results and Discussion

### 3.1. Silver Sulfide Stabilized by Chitosan

Synthesized Ag_2_S nanoparticles exhibited an average size of ≈3–5 nm as estimated by TEM micrographs (Appendix A) and a negative charge (measured by the Zetasizer Nano ZS), i.e., they were virtually free from Ag^+^ (Appendix A). The UV absorption spectrum showed a peaked shoulder at ≈300 nm, confirming the formation of nanoscale Ag_2_S in corroboration with TEM imaging. After their treatment with ascorbic acid with different concentrations, the Ag_2_S spectrum exhibited a surface plasmon peak at ~402 nm (Appendix A), indicating the generation of Ag nanoparticles on the Ag_2_S surface [16].

### 3.2. Morphological Characterization of the Modified GCE

The morphologies of Ag@AgCl/Ag_2_S located inside or outside of the CNTs were investigated by TEM imaging (Figure 1a). The TEM snapshots enabled an estimated 80% of spherical-shaped and tetrahedral of Ag@AgCl/Ag_2_S in the external channels of CNTs, whereas some Ag crystallites with long wire shapes were restricted in the gutters. SEM imaging of Ag@AgCl/Ag_2_S/CNTs on the GCE confirmed the presence of Ag@AgCl (10–40 nm in diameter) inside CNTs. Some larger Ag@AgCl NPs (≈100 nm in diameter), however, were located on the CNT walls and their ends (Figure 1b). The EDS spectra of Ag@AgCl/Ag_2_S/CNTs unveiled the presence of Ag together with a small Cl amount (Figure 1c), illustrating the deposition and dispersion of Ag@AgCl on CNTs in both the interior and surface of CNTs. No sulfur peak from EDS was observed, implying Ag_2_S NPs were completely covered by chitosan and Ag@AgCl NPs.

The formation of stabilized Ag_2_S NPs on the chitosan-CNTs composite was further confirmed by FTIR (Figure 1d). The absorption peaks at 1385 cm^−1^ and 1054 cm^−1^ were attributed to the O−H bending and the C−O stretching of CNTs [17]. The characteristic absorption peaks of −OH, −C=O and −C−O of the Ag@AgCl/Ag_2_S/CNTs composite significantly increased. The band at 3357–3290 cm^−1^ was assigned to ν (N−H) whereas two small peaks at 2931, and 2864 cm^−1^ were attributed to the −CH_2_ and −CH_3_ groups of chitosan [18]. The spectrum of Ag@AgCl/Ag_2_S/CNTs exhibited a peak at 1621 cm^−1^, reflecting the peak shifting of the C=O stretching (amide I) of chitosan, usually observed at ≈1650 cm^−1^. A decrease in the band intensity at ≈1550 cm^−1^ corresponding to the amino groups of chitosan suggested the mechanism involved in the stabilization of Ag_2_S and Ag@AgCl by chitosan was based on the interaction of primary −NH_2_ groups with the nanoparticle’s surface [19].

### 3.3. Electrochemical Properties of the Modified Electrode

For electrodeposition of Ag_2_S on CNTs/GCE, Ag and chitosan in Ag_2_S-chitosan can be oxidized and reduced on CNTs surface in the range of 0.1 V to −1.0 V, as shown in Appendix A [13]. The oxidation peak at ≈0.1 V assumed that it is Ag^0^ to Ag^+^ from Ag_2_S dispersed in chitosan [20]. The reduction peak at −0.4 to −0.9 V was the reduction of O_2_ and H_2_O. Electrodeposition of Ag@AgCl on the Ag_2_S/CNTs/GCE was also performed using the procedure described from our previous work [20] with the addition of 0.1 M KCl into the AgNO_3_ solution. The oxidation peak of ascorbic acid (peak II) with a small peak of AgCl oxidation (peak I) occurred at 0.55 V and 0.01 V, respectively (Figure 2a). The reverse scan showed a single reduction peak (peak III) at −0.11 V, which was attributed to the reduction of Ag^+^ to Ag^0^ and confirmed the presence of Ag@AgCl on Ag_2_S/CNTs [21]. The CV curve of the Ag@AgCl/Ag_2_S/CNTs/GCE (Figure 2b) exhibited a sharp oxidation current peak at 0.38 V (peak II) due to the oxidation of Ag^0^ to AgCl and its corresponding reduction current peak at −0.11 V (peak III, AgCl to Ag^0^) [22]. A weak oxidative current in the potential range of 0.27 V (peak I) was induced by an oxidative reaction of as-prepared carbon composites with oxygen-containing surface functional groups [23]. The reduction of dissolved O_2_ was confirmed by the reduction peak (peak IV) at −0.55 V [24].

Using the redox couple Fe(CN_6_)^3−/4−^ as the probe, the CNTs/GCE or the Ag@AgCl/CNTs/GCE yielded a higher current compared to the bare GCE or Ag_2_S/CNTs/GCE because of their higher electrical properties and higher active surface areas (Figure 2c). The Ag@AgCl/Ag_2_S/CNTs/GCE exhibited the smallest overpotential (ΔE_p_) together with the highest current, indicating the synergetic effects of heterogeneous catalysts. The high Cl^−^ concentration in the solution [25] also facilitated the oxidation/reduction of the Ag^0^ Ag^+^ pair (+0.1 V vs. −0.22 V).

The charge transfer resistance of the electrode (R_ct_), the most sensitive parameter, is denoted as a semicircle diameter in the Nyquist diagram. Nyquist plots with an equivalent circuit (Figure 2d) and pertinent estimated EIS data are summarized in Appendix A. The bare GCE displayed a semicircle (~386.8 Ω) sphere as predictable for a minimal electron transfer resistance toward the redox probe (Figure 2d). After modified by CNTs, the electrode became very electroactive as reflected by a small semicircle with R_ct_ = 29.33 Ω. These results attested to the existence of a high electron transfer pathway between the electrode-electrolyte boundary to allow the electron diffusion on the electrode surface. The semicircle of the Ag_2_S/CNTs/GCE was noticeably broader with R_ct_ of 87.17 Ω, confirming the presence of the Ag_2_S-chitosan semiconductor on CNTs. Finally, after being decorated with the Ag@AgCl, the Ag@AgCl/Ag_2_S/CNTs/GCE exhibited a pronounced decrease in the interfacial resistance with R_ct_ of ~46.38 Ω. Such results illustrated the resistive/capacitive behavior associated with electron transfer (faradaic process) coupled with a double layer charging process [26].

### 3.4. Effect of pH

The peak potentials of HQ and AR shifted towards more negative when the pH increased from 5.0 to 8.0, confirming the oxidation of the compounds involved in the proton-transfer process (Appendix A and Figure 3a). The relationship between H^+^ and electron is only valid in the acidic solution to neutral (pH 7.0), then becomes almost unchanged in the basic solution for most of the molecule which H^+^ relates to the redox reaction [27]. The electrochemical reaction becomes more difficult in basic solutions due to the shortage of protons [28]. The slope of the oxidation potential versus pH plot was −62.7 mV⋅pH^−1^ for HQ and −57.2 mV⋅pH^−1^ for AR, respectively. Such values were close to the theoretical value of −59 (h/n) mV⋅pH^−1^ for a redox reaction, where n (number of electrons) is equal to h (number of protons) [29]. However, a smaller effect was observed in AA2G (Appendix Ac and Figure 3a). The oxidation potential versus pH (5 to 8) plot exhibited a slope of only −22.5 mV⋅pH^−1^ for AA2G, implying the oxidation was difficult and became less sensitive to pH due to the lack of proton.

The pH effect on the anodic peak current of all analytes in DPV mode was also studied and shown in Figure 3b,c. Notice the anodic peak currents after the blank subtraction of HQ and AR were highest at pH 6–7, significantly reduced in basic media (Figure 3c). In contrast, the peak current of AA2G was increased with increasing of pH from 5 to 8. It was consistent with the adsorption of AA2G on chitosan and CNTs at high pH [30]. Consequently, pH 7.0 was selected for all subsequent measurements as a compromise between detection sensitivity and selectivity.

### 3.5. Electrocatalysis of HQ, AR, and AA2G at the Ag@AgCl/Ag_2_S/CNTs/GCE

The oxidation reaction mechanisms of HQ, AR, and AA2G on the Ag@AgCl/Ag_2_S/CNTs/GCE were studied by plotting the CV behavior at various. scan rates (ν). Among the three analytes, a reversible redox reaction was only observed for HQ; therefore, it was used to evaluate the reversible electron transfer phenomenon (Figure 4). In this case, the peak current (I_p_) versus ν was linear with R^2^ > 0.99, signifying the dominance of the adsorption process [31].
$$E_{p}=E^{\emptyset }+\frac{2.303RT}{\left(1-\alpha \right)nF}log\;v$$

The cathodic signals for AR and AA2G during the reverse scan in the potentials range of −0.8 to 1.0 V not only stemmed from HQ and ascorbic but were also related to a glucose moiety of AA2G or AR (Appendix A). The synergistic effect of Ag@AgCl, Ag_2_S, chitosan, and CNTs contributed to increased agility in measurements and reduced the oxidation potential of the analytes through π–π stacking, electrostatic, and hydrogen bonding.

### 3.6. Analytical Performance of the Modified Electrode

DPV was conducted to investigate the electrochemical behavior of the Ag@AgCl/CNTs/GCE and Ag@AgCl/AgS/CNTs/GCE for 9.1 μM HQ, 3.7 μM AR and 3.0 μM AA2G (Figure 5a). As expected, the Ag@AgCl/AgS/CNTs/GCE and the Ag@AgCl/CNTs/GCE exhibited anodic peaks of HQ, AR, or AA2G. However, the former peak currents were about 5-fold higher than those of the latter, indicating the assisted electron transfer of Ag_2_S and chitosan. The HQ anodic peak potential also shifted from 200 mV to 79 mV when the combined Ag@AgCl and Ag_2_S were used to modify the CNTs/GCE (Figure 5a). The AR and AA2G peaks were not observed by the CNTs/GCE or bare GCE. After the CNTs/GCE were modified by Ag@AgCl or Ag_2_S-chitosan, both AR and AA2G were detected; however, Ag@AgCl/CNTs were much more active than Ag_2_S/CNTs. The oxidation of Ag to Ag^+^ at 0.28 V disappeared after the first run (Figure 5b) and even after running CVs in the 0.1 M of the KCl solution with 10 cycles. Such a result was attributed to the reduction of Ag^+^ to Ag^0^ on Ag@AgCl by HQ [32]. However, the oxidation peak at 0.28 V was recovered after the modified electrode was left overnight at ambient temperature, implying the photo-induced generation of AgNPs, AgCl, and Ag_2_S [11].

A typical DPV of the Ag@AgCl/Ag_2_S/CNTs/GCE with successive addition of a standard analyte solution shown in Figure 5c. A well-defined DPV response of each analyte was observed, demonstrating stable and efficient catalysis of Ag@AgCl/Ag_2_S/CNTs. The anodic current was linear with the concentration from submicromolar to micromolar levels with a correlation coefficient of >0.99 (Figure 5d and Table 1). The detection limit (LOD at S/N = 3; *n* = 3) was determined as 0.41 (±0.02) μM for HQ, 0.12 (±0.01) μM for AR, and 0.25 (±0.01) μM for AA2G. The electrochemical parameters of the Ag@AgCl/Ag_2_S/CNTs modified GCE were compared with other electrochemical sensors. The performance of Ag@AgCl/Ag_2_S/CNTs modified GCE is comparable to other modified electrodes as shown in Table 1.

The signal response by varying one analyte concentration at a fixed concentration of the others was investigated (Appendix A). When only the HQ concentration was varied, the I_pa_ of the others was not significantly changed. However, increasing AR or AA2G concentration significantly affected the HQ signal and the AR signal, respectively. This behavior might be attributed to the competitive diffusion of small molecules and the larger molecules on the electrode surface [34]. Repeatability for the simultaneous analysis of 9.1 μM HQ, 3.7 μM AR, and 3.0 μM AA2G was studied. The relative standard deviation (RSD) of the anodic peak current was 6.2% for HQ, 6.3% for AR, and 5.8% for AA2G. Three different prepared electrodes were used to measure the standard solutions at the same concentration to assess the fabrication reproducibility (Appendix A). The relative standard deviations (RSDs) are 6.30%, 2.94%, and 2.28% for HQ, AR, and AA2G, respectively. The storage stability of the Ag@AgCl/Ag_2_S/CNTs/GCE was also examined under ambient temperature (Appendix A). The response signals of AR and AA2G were diminished while the signal of HQ increased, indicating the adsorption of HQ onto the electrode surface [32]. Some commercial whitening products might contain ascorbic acid and kojic acid together with AR and AA2G. Therefore, the plausible interfering effects of these two acids were also investigated. In the presence of kojic acid (281.5 μM) and ascorbic acid (227.1 μM), i.e., more than 20-times higher than the analyte concentration, the current responses of HQ and AA2G remained almost unchanged (less than ±5%), as shown in Appendix A.

### 3.7. Real Sample Analysis

The modified electrode was then conducted to detect HQ, AR, and AA2G in a commercial whitening lotion (0.5 mL of lotion sample in 5.00 mL of 0.1 M phosphate buffer, pH 7). The standard solution of all three analytes was spiked into the lotion samples to estimate the recovery for the analytes. The mixtures were sonicated for 15 min before the analysis was performed by DPV. From spiking 9.1 μM HQ, 3.7 μM AR, and 3.0 μM AA2G standard solution, the estimated analyte recovery in the lotion sample ranged from 94.8 to 106.3%: HQ = 94.8% (±8.1), AR = 105.9% (±1.8), and AA2G = 106.3% (±5.4). The relative standard deviations (RSD) of 1.83% and 8.07% illustrated the good repeatability of the modified sensor.

The standard addition method (3 replicates, Appendix A) was performed to probe the signal response when a mixture of HQ, AR, and AA2G was spiked into the whitening lotion and compared with the HPLC method. The conditions and sample preparation for the HPLC method were described in a previous report with some modifications [3]. The response signal obtained from HQ was subtracted by the electrolyte background signal for the DPV method. The AR and AA2G peaks shifted from 0.47 V to 0.4V and 0.77 V to 0.53 V, respectively. This phenomenon was possibly due to the adsorption of emulsifier molecules in the lotion sample on the electrode surface [35]. The concentration of all analytes from the standard addition method is shown in Table 2. The AR concentration in lotion samples was very high while the HQ content could not be detected by these two methods. Such results confirmed the sensor’s applicability for detecting the decomposition of HQ derivatives in the lotion sample without interference from ascorbic acid and AA2G. The matrix effect was also studied by comparing the slope of a calibration curve of three standard solutions with that of the corresponding matrix-spiked standard solutions (Appendix A and Appendix A). According to the percentage of the matrix effect (%ME), this method was affected by the ions in the matrix, which suppressed the response signals (%ME < 100). Hence, the sample preparation step should be performed with a pertinent dilution factor for each analyte together with an appropriate sample cleaning procedure to circumvent the matrix effect.

## 4. Conclusions

This work unraveled a simple strategy for the fabrication of the Ag@AgCl/Ag_2_S/CNTs modified GCE to exploit the synergetic effect of Ag@AgCl NPs, Ag_2_S NPs, chitosan, and CNTs. Chitosan played an important role to stabilize Ag_2_S and interact with the glucose moiety of AR and AA2G, two popular ingredients of commercial whitening lotion products. With high detection sensitivity, this modified electrode could be extended for analysis of ascorbic acid and other electroactive compounds in food, cosmetics, and other horticultural products.

## Figures and Tables

**Figure 1 nanomaterials-10-01583-f001:**
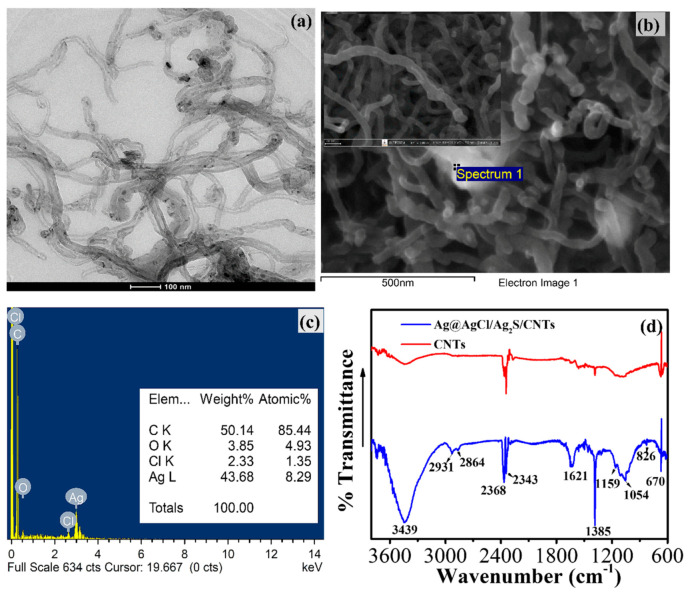
(**a**) TEM image of Ag@AgCl/Ag_2_S/CNTs; (**b**) SEM images of Ag@AgCl/Ag_2_S/CNTs at low-magnification and higher-magnification (inset); (**c**) with the energy dispersive X-ray (EDS) spectrum, (**d**) FTIR spectrum of Ag@AgCl/Ag_2_S/CNTs.

**Figure 2 nanomaterials-10-01583-f002:**
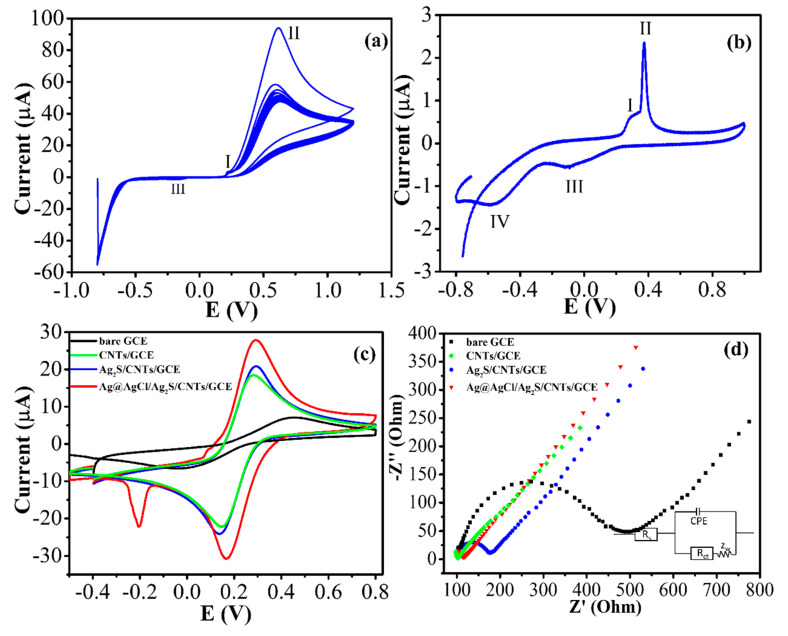
(**a**) Cyclic voltammograms (CVs) of the Ag_2_S/CNTs/GCE in 0.1 M KCl containing 1% (*w/v*) ascorbic acid and 5 mM of AgNO_3_ at 20 mV⋅s^−1^; (**b**) CVs of the Ag@AgCl/Ag_2_S/CNTs/GCE in 0.1 M phosphate buffer, pH 7 at 100 mV⋅s^−1^; (**c**) CVs obtained by the bare GCE, and the modified electrode in 0.1 M KCl containing 5 mM [Fe(CN)_6_]^3–/4^; (**d**) EIS curves of the bare GCE, and the modified electrode in 0.1 M KCl containing 5 mM [Fe(CN)_6_]^3–/4−^. Amplitude: 5 mV, Frequency: 0.1 Hz to 1000 kHz

**Figure 3 nanomaterials-10-01583-f003:**
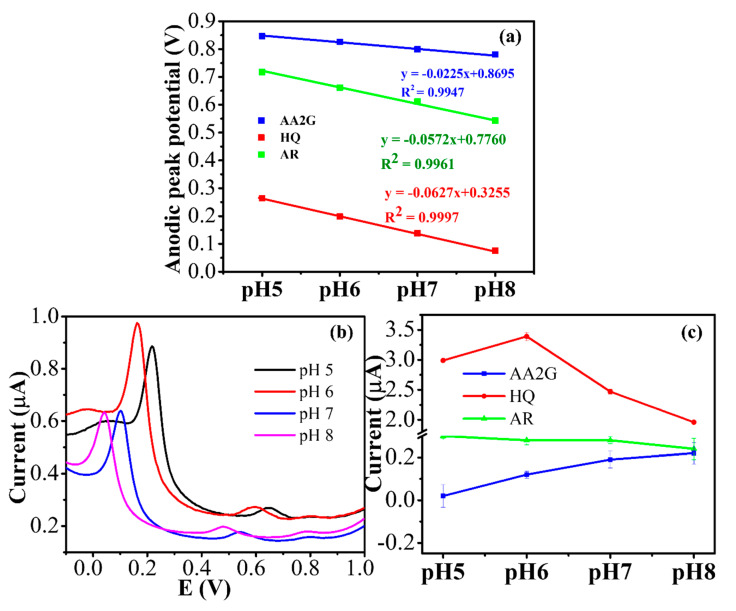
(**a**) A plot of the anodic peak potential of all analytes vs. pHs; (**b**) DPV signals of the HQ (4.54 μM), AR (1.84 μM) and AA2G (1.48 μM) determination in the phosphate buffer solution at different pH values; (**c**) Effect of pH on the peak current of the analytes.

**Figure 4 nanomaterials-10-01583-f004:**
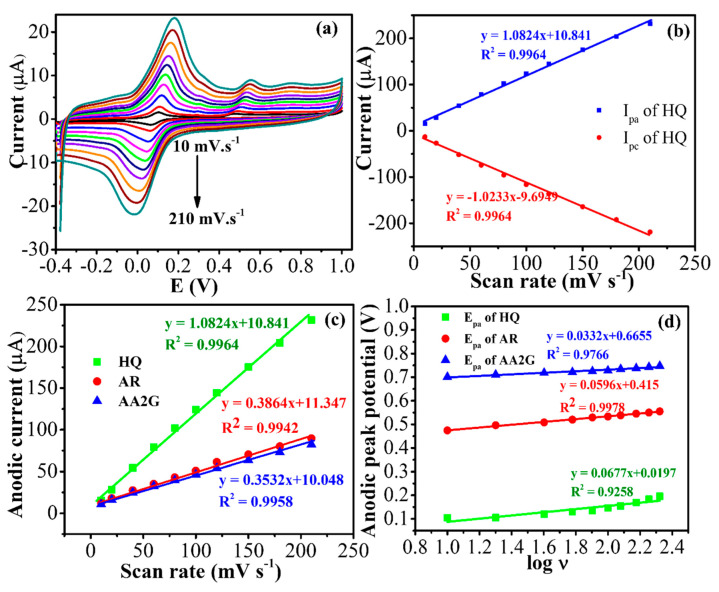
(**a**) CVs of the Ag@AgCl/Ag_2_S/CNTs/GCE in phosphate buffer, pH 7 containing HQ (18.2 μM), AR (7.35 μM), and AA2G (5.91 μM) of all analytes at different scan rates: 10 to 210 mV⋅s^−1^; (**b**) the peak current versus the scan rate; (**c**) the anodic peak current versus the scan rate; (**d**) the anodic peak potential versus the logarithm of the scan rate (X-axis).

**Figure 5 nanomaterials-10-01583-f005:**
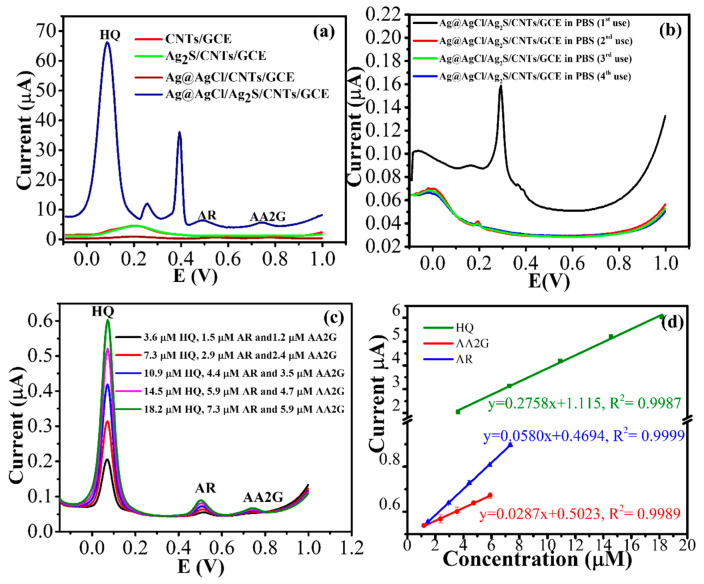
(**a**) DPVs obtained at CNTs/GCE, Ag_2_S/CNTs/GCE, Ag@AgCl/CNTs/GCE, and Ag@AgCl/Ag_2_S/CNTs/GCE in 0.1 M phosphate buffer (pH 7.0) containing 9.1 μM HQ, 3.7 μM AR, and 3.0 μM AA2G; (**b**) DPVs obtained at the Ag@AgCl/Ag_2_S/CNTs/GCE in 0.1 M phosphate buffer, pH 7.0 after 10 cycles in the 0.1 M KCl solution; (**c**) DPVs of the Ag@AgCl/Ag_2_S/CNTs/GCE vs. Ag/AgCl/KCl upon successive determination of different analytes concentrations in 0.1 M phosphate buffer; (**d**) Anodic current responses versus the analyte concentration.

**Table 1 nanomaterials-10-01583-t001:** Comparison of the performance of the proposed method with the previously reported sensor.

Electrode	Analyte	LOD (µM)	Potential (V)	Linear Range (μM)	Method	References
Carbon/ferrocene paste	HQ	0.06	0.213	0.20–10	LSV	[4]
	AR	-	0.660			
Cu PCB/SPE ^1^	AA2G	0.73	0.6	7.39–473	CV	[5]
ERGO-pEBT/AuNPs ^2^	HQ	0.015	0.166	0.52–31.4	DPV	[7]
	CC ^4^	0.008	0.277	1.44–31.2		
	RC ^5^	0.039	0.660	3.8–72.2		
HAP-ZnO-Pd NPs/CPE ^3^	AR	0.086	0.50	0.12–56	DPV	[33]
	VC ^6^	0.019	0.7	0.12–55.36		
Ag@AgCl/Ag_2_S/CNTs/GCE	HQ	0.4	0.05	0.91–27.2	DPV	This work
	AR	0.1	0.5	0.73–14.7		
	AA2G	0.25	0.7	1.18–11.8		

^1^ Copper-enriched printed-circuit-board waste; ^2^ reduced graphene oxide, poly (Eriochrome black T), and gold nanoparticles; ^3^ hydroxyapatite-ZnO-Pd NPs modified carbon paste electrode; ^4^ catechol; ^5^ resorcinol; ^6^ ascorbic acid.

**Table 2 nanomaterials-10-01583-t002:** The content of hydroquinone, arbutin, and ascorbyl glucoside in the whitening lotion (*n* =3).

	Concentration Found (mg⋅L^−1^) HPLC	Concentration Found (mg⋅L^−1^) DPV
HQ	ND	ND
AR	2.1 (±0.3) × 10^4^	2.1 (±0.1) × 10^4^
AA2G	12.5 (±1.5)	40.2 (±0.7)

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
