# Peer review of "Simultaneous Analysis of Hydroquinone, Arbutin, and Ascorbyl Glucoside Using a Nanocomposite of Ag@AgCl Nanoparticles, Ag2S Nanoparticles, Multiwall Carbon Nanotubes, and Chitosan"

_nanomaterials, 2020, doi:10.3390/nano10081583_

Round 1
Reviewer 1 Report
Dear Editor,
This paper described an electrochemical sensor based on nanocomposite comprising Ag nanoparticles on AgCl/Ag2S nanoparticles was decorated on multi-walled carbon nanotubes and used to modify a glassy carbon electrode for the detection of hydroquinone (HQ), arbutin (AR) and ascorbyl glucoside (AA2G). Authors presented results about the performance of this sensor. Although these data are interesting, I do not consider that experimental results shown are enough for publishing this paper. The Authors should present some additional experimental results.
Moreover, the paper is well structured. Therefore, I do recommend considering the publication of this paper after addressing the following major concerns.
- Some analytical characteristics of the sensor should be indicated in the abstract, e. g. lineal range, LOD, LOQ etc.
- The Authors should describe the protocol was used for the carboxylation carbon nanotubes.
- 4. Fabrication of the Ag@AgCl/Ag2S/CNTs/GCE: The Authors should describe the reactions involved in the immobilization of the Ag2S colloidal solution on the CNTs/GCE
- Figure 2a: I do not understand why in the measure solution the Authors include in the mixture AgNO3, please explain it.
- The Authors should include error bars in the Figures.
- The Authors should indicate how many numbers of the replicates was used for calculated the LOD, LOQ and RSD.
- The Authors should indicate the interval range of the analytes.
- The Authors should explain how develop the stability studies (with only one electrode, where and the storage conditions of the sensor…
- The Authors should include studies of the reproducibility and repeatability.
- The Authors should show the samples results in a table
- The Authors should calculate the statistical comparison of the slopes for the study of matrix effect.
Sincerely,
The reviewer
Author Response
Dear Reviewer
Thank you for the suggestion to complete our work. Your suggestions are very valuable to us. We made some revisions and corrections as you suggested and attached the revision details in the attached file below. Please see the attached file.
Regards
The authors

Reviewer 2 Report
Dear Editor,
In the present work Authors report the realization of a “glassy carbon electrode modified by a composite of Ag@AgCl, Ag2S, CNTs, and chitosan for the simultaneous analysis of HQ, AR, and AA2G". In this respect, the paper is new and the topic is certainly of interest to the readers of the "Nanomaterials" Journal.
However, the work needs to be deeply revised as it has several weaknesses that need to be resolved to be suitable for publication. Particularly critical is the characterization of the composite material, both in terms of morphology and electrochemical properties.
Some specific critical points to be adressed:
Figure S1 (Left): In this image, it is very difficult to find structure of 600 nm as stated from Authors. Please comment!
Figure S1 (Right): These data are not conclusive. It is difficult to find a correlation between UV traces and chitosan concentrations.
Figure S3: The peak at 402nm has been discussed in the paper. Please add also a comment to explain the peak at 250 nm.
Figure 3. (B) "The oxidation potential of HQ 194 (4.54 M), AR (1.84 M) and AA2G (1.48 M) in the DPV mode with different pH values ranging from 4.0–8.0": in this image there are only 4 traces! (C) "Dependence of the anodic peak current (Ipa) with pHs": is not possible to correlate Ip vs pH as a consequence of fihgure 3 (B). Please correct.
Paragraph: “3.2. Morphological characterization of the modified GCE”. The present paragraph is difficult to be understood. Data and comment are reported in a very confused mode. I really cannot find the structure suggested by the authors based on the pictures reported in Figure 1. In addition, please clarify the presence of Ag@AgCl/MWCNTs! Again, how does IR data confirm the presence of Ag2S NPs structures in chitosan?
Figure 6. Data reported in this figure are redoundant. Please delete B and C panels.
Definitely, in my opinion the paper needs to be greatly improved, to be published.
Author Response
Dear Reviewer
Thank you for the suggestion of the complete work adjustment. Your suggestions are very valuable to us. We made some revisions as you suggested and attached the revision details in the attached file below.
Regards
The authors

Reviewer 3 Report
The manuscript describes the development of an electrochemical sensor for the simultaneous analysis of hydroquinone, arbutin and ascorbyl glucoside using a nanocomposite of Ag@AgCl nanoparticles, Ag2S nanoparticles, multi-walled carbon nanotubes, and chitosan. The studies were performed well and described with sound discussions. however, there are a couple of issues that must be resolved in a major revision before a decision upon publication can be made.
- The authors must correct the labelling of x-axes in all their voltammograms. They should replace "Potential applied (V)" with "E (V) vs. Ag/AgCl" according to the reference electrode that they utilized.
- The authors must compare the analytical performance of their sensor with the existing sensors or other detection technologies (such as HPLC) in literature for the same target analytes. This information must be presented in a Table that includes the nanomaterials used in the sensors with detection limits and dynamic linear range of each analyte.
- The authors must also show the equivalent circuit that can be applied to analyze the Nyquist plots shown in Figure 2D. did the authors run a simulation to fir the data in Nyquist plots? What do the lines that connect the dots mean? Are those lines indicating the results of a simulation to fit the data into the equivalent circuit?
In view of my comments above, I would recommend a major revision of the manuscript.
Author Response
Dear Reviewer
Thank you for the suggestion to complete the work adjustment. Your suggestions are very valuable to us. We made some adjustments as you suggested and attached the revision details in the attached file below.
Regards
The author

Round 2
Reviewer 1 Report
Dear Editor,
This paper described an electrochemical sensor based on nanocomposite comprising Ag nanoparticles on AgCl/Ag2S nanoparticles was decorated on multi-walled carbon nanotubes and used to modify a glassy carbon electrode for the detection of hydroquinone (HQ), arbutin (AR) and ascorbyl glucoside (AA2G). Authors presented results about the performance of this sensor. Although these data are interesting, I do not consider that experimental results shown are enough for publishing this paper. The Authors should present some additional experimental results.
Moreover, the paper is well structured. Therefore, I do recommend considering the publication of this paper after addressing the following major concerns.
- 4. Fabrication of the Ag@AgCl/Ag2S/CNTs/GCE: The Authors should describe the reactions involved in the immobilization of the Ag2S colloidal solution on the CNTs/GCE
We have tried to modified Ag2S-chitosan by drop-casting method but the layer was too thick. According to the literatures, the metal complexchitosan can be deposited on the electrode by applying the low voltage.
In our work, Ag in Ag2S can be oxidized and reduced on CNTs surface in the range of 0.1 V to -1.0 V. We got the oxidation peak at around 0-0.1 V which we assumed that it Ag0 Ag+ from Ag2S. We followed the ref. 13 (Geng, Z.; Wang, X.; Guo, X.; Zhang, Z.; Chen, Y.; Wang, Y. Electrodeposition of chitosan based on coordination with metal ions in situ-generated by electrochemical oxidation. J. Mater. Chem. B 2016, 4, 3331–3338.). We did not mention in our work because we think it was not the highlight of our work.
The Authors should include the explanation in the paper.
- The Authors should calculate the statistical comparison of the slopes for the study of matrix effect.
In general, the electrochemical sensor will have a high sensitivity at the low concentration of the analyte. The sensitive of detection was reduced after determination the analyte at high concentration and complex matric environment sample. These problems can be solved by diluted the sample to very low concentration. Therefore, the slope of calibration curve that we have got from the determination was similar.
I do not understand, the slopes are different! For example, 0.0251 vs. 7.94E-9, this results show evident matrix effect and what is the dilution used in this work? and how did the Authors calculated the results showed in this work in the sample analisis?
Sincerely,
The reviewer
Author Response
"Please see the attachment."

Reviewer 2 Report
Dear Editor, this paper is the rvised version of a manuscript alredy propsed from the authors.
However, although authors have clarified and corrected some of the critical points, some revisions are still needed.
For example, figure 3B has not been correct (or in any case it is not consistent with the caption and the related comment in the text).
In Figure S6 D it is noted that the y axis of the currents are in mA. This is impossible considering that all reported processes are related to currents of the order of micro-A.
Furthermore, I still have a doubt. How come you never see the presence of Ag2S NPs in the experiments? The fact that it is covered by Ag @ AgCl, considering that these are nano-sized, does not exclude the possibility of being viewed with EDS spectroscopy (generally capable of analyzing thicknesses of the order of a few mm).
Please add a comment.
Author Response
"Please see the attachment."

Reviewer 3 Report
The authors seem to have prepared the revised version of their manuscript inn a rush, not paid enough attention to the presentation of their figures. I would recommend the authors to be extra careful about the visibility of their figures and insets so that the readers can easily download or print out the article and read on their screen or on paper.
For example, the authors must correct the inset in Figure 2D. The equivalent circuit of the Nyquist plot is impossible to see in the inset of Figure 2D. When I try to zoom in the inset on my computer screen, the equivalent circuit vaguely seems very different than the general Randles circuit. The authors must explain clearly why they identified such a unique circuit. what was the error range in terms of the elements of the circuit such as charge transfer resistance, double-layer capacitance, etc? These numerical values should have been presented a Table depicting their changes upon the immobilization of nanoparticle layers on the electrode surface. This table should have been presented in the Supplementary Information file.
The authors should have also corrected the y-axis of Figure 5A because the current unit seems to be milliAmpere, where it should have been microAmpere. In the x-axes of Figure S4A, B and C and Figure S5A, B, C and D, and Figure S9A and B, the authors should have replaced Potential applied (V) with E (V) to be in agreement with the rest of Figures in the article. In Figure S6D, the y-axis current unit seems to be milliAmpere, where it should have been microAmpere. In Figure S8, the Current unit seems to be milliMolar in y-axis, which should be corrected as microAmpere. Figure S8 caption also needs more explanation to describe what is displayed in panels A and B. There are no labels of A and B in that figure. In Figure S10, are the authors sure that the unit of concentration in Figure SD is milligram per Liter instead of microgram per Liter? The unit seems to be microgram per Liter in Figure S10 B.
In view of my comments above, I have to recommend the authors to revise their manuscript carefully in a major revision.
Author Response
"Please see the attachment."
